

# Angle information assisting skeleton-based actions recognition

Chengming Liu[1], Jiahao Guan[1], Haibo Pang[1], Lei Shi[1] and Yidan Chen[2]

[1] School of Cyber Science and Engineering, Zhengzhou University, Zhengzhou, Henan, China
[2] National Engineering Laboratory for Cyber Science and Technology, Shangu Cyber Security Technology Company Limited, Zhengzhou, Henan, China

## ABSTRACT

In human skeleton-based action recognition, graph convolutional networks (GCN) have shown significant success. However, existing state-of-the-art methods struggle with complex actions, such as figure skating, where performance is often unsatisfactory. This issue arises from two main factors: the lack of shift, scale, and rotation invariance in GCN, making them especially vulnerable to perspective distortions in 2D coordinates, and the high variability in displacement velocity, which depends more on the athlete's individual capabilities than the actions themselves, reducing the effectiveness of motion information. To address these challenges, we propose a novel cosine stream to enhance the robustness of spatial features and introduce a Keyframe Sampling algorithm for more effective temporal feature extraction, eliminating the need for motion information. Our methods do not require modifications to the backbone. Experiments on the FSD-10, FineGYM, and NTU RGB+D datasets demonstrate a 2.6% improvement in Top-1 accuracy on the FSD-10 figure skating dataset compared to current state-of-the-art methods. The code has been made available at: https://github.com/Jiahao-Guan/pyskl_cosine.

# INTRODUCTION

Action recognition has become an active research area in recent years, as it plays a significant role in video understanding. Prior investigations have explored various modalities for feature representation, such as RGB frames, optical flows, audio waves, and human skeletons. Among these modalities, skeleton-based action recognition has garnered heightened interest in recent years due to its action-focusing nature and robustness against complicated background. Early pioneering efforts primarily relied on recurrent neural networks (RNN) (*Du, Wang & Wang, 2015*; *Song et al., 2017*) and convolutional neural networks (CNN) (*Ke et al., 2017*; *Liu, Liu & Chen, 2017*) to extract features or generate pseudo-images from human joint data for action recognition. While these methods achieved reasonable performance, they were fundamentally limited in their ability to model the complex inter-dependencies between joints, a crucial aspect for fine-grained action recognition.

Graph convolutional networks (GCN) have become a popular approach for skeleton-based action recognition. *Yan, Xiong & Lin (2018)* were among the first to apply GCN with temporal convolution for this task. However, their model uses a fixed, pre-defined

Corresponding author
Haibo Pang, panghbzzu@163.com

topology based on the physical structure of the human skeleton, which limits its expressive power across different network layers.

To bolster the capabilities of GCN, recent approaches (*Shi et al., 2019*, *2020*; *Liu et al., 2020b*; *Chen et al., 2021*) have aimed to acquire more fitting topologies. They successfully transformed the skeleton topology into a learnable structure by introducing a learnable weight matrix and combining it with the skeleton topology, enabling the model to learn the dependency relationship between two physically unrelated joint points, such as two hands. However, GCN-based approaches struggle with shift and scale invariance. Specifically, when working with 2D data, projecting from 3D to 2D can change the apparent size of the characters in the image, with objects closer appearing larger and those farther away appearing smaller, as illustrated in Fig. 1 right side. Additionally, horizontal movement can alter positions, and these changes in scale and position may not reflect the actual movement, especially for sports with large displacements, like figure skating. When the model lacks shift, scale invariance, these coordinate changes can interfere with the model's recognition.

To solve above problems, In this work, We introduce a new angle feature that complements existing joint and bone features. Additionally, we present an improved keyframe Sampling algorithm that accounts for sample randomness and better preserves important semantic information in keyframes.

Drawing inspiration from leveraging high-order information, we leverage novel high-order information extracted from skeleton data to quantify the joint range of motion between two bones. The joint range of motion, typically measured in degrees, provides a valuable metric for assessing joint flexibility and mobility. Inherently linked to posture and movement, it emerges as an inherently discriminative feature for action recognition tasks. We represent the joint range of motion through cosine similarities between pairs of bone vectors, forming the foundation of our cosine stream. By feeding these cosine similarities into a graph convolutional network, we make predictions for action labels. Simultaneously, the cosine stream integrates with the joint-bone two-stream network, giving rise to the development of a comprehensive three-stream network.

In sampling, a recent work by *Duan et al. (2022a)* introduces a uniform sampling technique, evenly dividing sequences into N non-overlapping segments with an equal number of frames. One frame is then randomly selected from each segment and aggregated to form a new sub-sequence. While effective, this method overlooks considerations for keyframes. Building upon ideas from *Liu et al. (2020b)* and *Wang et al. (2019)*, we enhance the approach by simplifying the keyframe selection strategy and integrating it with the original uniform sampling. Various combination strategies have been explored, resulting in particularly significant improvements in the joint stream and bone stream.

Our contribution are summarized as follows:

- We propose a cosine stream, which quantifies the joint range of motion between two bones in degrees, to assist in action recognition with significant displacement.
- We have enhanced the existing downsampling algorithm by integrating the keyframe concept. This enhancement yields substantial improvements in both the joint and bone streams.

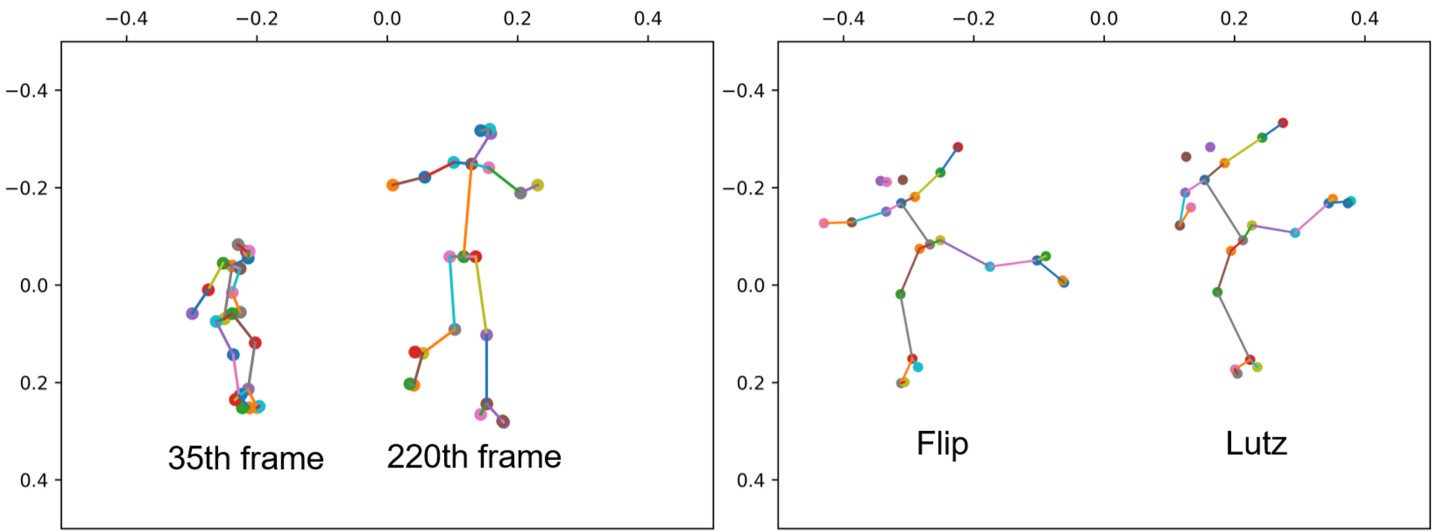

**Figure 1 The two people in the left image correspond to the 35th and 220th frames of the same sample human skeleton, respectively.** The two people in the image on the right correspond to the take off stage of Flip jump and the take off stage of Lutz jump in figure skating, respectively.

- The experimental results indicate that our method can enhance the accuracy performance of the model without necessitating modifications to the network structure itself.

## RELATED WORK

GCN (*Bruna et al., 2014*; *Defferrard, Bresson & Vandergheynst, 2016*; *Kipf & Welling, 2016*; *Niepert, Ahmed & Kutzkov, 2016*; *Velickovic et al., 2017*; *Duvenaud et al., 2015*) is widely adopted in skeleton-based action recognition. It models human skeleton sequences as spatiotemporal graphs. *Yan, Xiong & Lin (2018)* introduced spatial temporal graph convolutional networks (ST-GCN), a widely recognized baseline for GCN-based approaches. ST-GCN integrates spatial graph convolutions and temporal convolutions to model spatiotemporal data, but it uses a static and fixed skeleton topology.

*Shi et al. (2019)* addressed this limitation by incorporating an adaptive graph topology and proposing the adaptive graph convolutional networks (AGCN). AGCN integrates a bone stream with a joint stream in a two-stream network. In the joint stream, the coordinate vector of each joint serves as an attribute for the corresponding vertex. For 2D coordinates, the first two channels represent the x and y coordinates of each joint, while the third channel conveys the overall confidence score of the coordinates, indicating the reliability of the joint positions. For 3D coordinates, the three channels represent the x, y, and z coordinates of each joint. In the bone stream, edges in the graph structure are treated as directed edges pointing from the body's periphery toward the center. Here, each vertex stores outgoing edge vectors rather than its own position, with the central vertex having no outgoing edges and thus assigned a zero vector. Additionally, *Shi et al. (2020)* proposed extracting motion information by computing the coordinate differences of joints and

bones between consecutive frames, which are then combined into a multi-stream network. These works propose adaptive topology structures that are data-driven and can learn the most suitable topology based on data, exploring various features to form multi-stream networks. However, the ability to capture dependencies between non-adjacent joints that are far apart in the skeleton remains limited.

Liu et al. (2020b) introduced a disentangled multi-scale aggregation scheme to remove redundant dependencies between vertex features from various neighborhoods. They also introduced a three-dimensional graph convolution operator that facilitates direct information flow across space and time, though this operator can be computationally intensive. Chen et al. (2021) proposed a Channel-wise Topology Refinement Graph Convolution Network (CTR-GCN), which dynamically models channel-wise topology in a refinement approach. This leads to flexible and effective correlation modeling and is also a strong baseline.

Based on AGCN and CTR-GCN, InfoGCN (Chi et al., 2022) and HD-GCN (Lee et al., 2023) have been proposed. InfoGCN introduces the concept of information bottleneck from information theory into this field and proposes a self-attention module. HD-GCN, on the other hand, presents a hierarchically decomposed graph to better identify significant distant edges within the same hierarchical subsets. It also includes an attention-guided hierarchy aggregation module that emphasizes key hierarchical edge sets through representative spatial average pooling and hierarchical edge convolution. Additionally, HD-GCN uses different center points from the human skeleton to design a six-way ensemble method for skeleton-based action recognition. However, both InfoGCN and HD-GCN are modifications of AGCN and CTR-GCN, incorporating numerous attention-based modules. As a result, they require large amounts of data and are prone to overfitting when data is insufficient.

The works most similar to ours are Hou et al. (2022) and Qin et al. (2022), both of which utilize angular features. Hou et al. (2022) proposed a dynamic anchor-based angle calculation method, which computes the angle between vectors formed by two bone points and an anchor point. This approach requires a substantial amount of training data. In contrast, Qin et al. (2022) employs a similar angle calculation method to ours.They focuses on exploring various angle calculation methods in the spatial dimension, whereas our research extends into the temporal dimension. We investigate the role of angular features in downsampling algorithms and propose a Keyframe Sampling algorithm.

## METHOD

### Cosine stream

In figure skating, athletes often maintain a consistent skating speed during the execution of actions, leading to significant displacement and variations in position features. The angle of the ice skate blade relative to the ice surface, distinguishing between the inside and outside edges, is a crucial factor in action classification. Figure 1 left side showcases two similar jumps in figure skating: the Flip and the Lutz. In the Flip jump on the left, the skater positions the left foot on the inner edge of the skate blade, shifting the overall body weight towards the inside of the blade, while extending the left arm naturally. In contrast, for the

Lutz jump on the right, the only difference lies in the skater placing the left foot on the outer edge of the skate blade, resulting in a relatively outward shift of the body weight. To achieve stability and enhance takeoff power, skaters typically opt to naturally curve their left hand towards the right side. To preserve blade clarity, slight differences in body joint angles occur, which tend to remain relatively stable during displacement compared to coordinates. Joint angle changes are typically induced by specific actions, making them more discriminative.

We aim to input human body joint angles as raw features into the network in the form of cosine similarities. The cosine similarity $\cos v_i$ for vertex $v_i$ is calculated using Eq. 1, where $\mathcal{N}(v_i)$ is the neighborhood of $v_i$, $A^2_{|\mathcal{N}(v_i)|}$ is the number of permutations, and $\overrightarrow{e_{ij}}$ is the vector from $v_i$ to $v_j$. Values for vertices in the cosine stream graph are generated, and an empty cosine similarity with a value of 0 is added to the outermost vertices, ensuring consistency in the design of the graph and network of cosine with that of joints and bones. Due to consistency in data format, our network can be easily integrated with mainstream joint bone two stream networks, as shown in the Fig. 2.

$$\cos v_i = \frac{1}{A^2_{|\mathcal{N}(v_i)|}} \sum_{j \in \mathcal{N}(v_i)} \sum_{k \in \{\mathcal{N}(v_i) - j\}} \frac{\overrightarrow{e_{ij}}}{||\overrightarrow{e_{ij}}||} \frac{\overrightarrow{e_{ik}}}{||\overrightarrow{e_{ik}}||}. \tag{1}$$

Regarding other calculation methods for joint angles, further exploration has been conducted in *Qin et al. (2022)*, and we will not provide too much repetitive introduction here.

## Keyframe sampling algorithm

Sampling keyframes is a crucial aspect of video analysis in figure skating, ensuring that selected frames encapsulate the most discriminative information within a video. In the figure skating task, the fast changing motion frames are distinctly important for jump action. Through Eq. (1), we transformed the original joint stream into a cosine stream, with attribute values stored in the vertices ranging from −1 to 1. This reflects the joint's range of motion from 180 degrees to 0 degrees, eliminating the need for normalization. We then sum the angles for all joints in the body to obtain the downsampling indicator, representing the extent of limb extension in each frame. A smaller value indicates a larger sum of angles for various joints in the entire body, implying greater limb extension. Hence, it is immensely beneficial in identifying keyframes within sequences. For instance, in a jumping sequence, the indicator during the takeoff phase is stronger than that during the mid-air spinning phase. This suggests that recognizing the takeoff is more crucial than identifying the posture during mid-air action, aligning with the focus on judging actions in figure skating sports.

We decided to incorporate uniform sampling, as introduced by *Duan et al. (2022c, 2022b, 2022a)*, into our proposed keyframe sampling approach, leading to various fusion strategies:

(0) Create a sequence of N+M frames using uniform sampling as the control sequence.

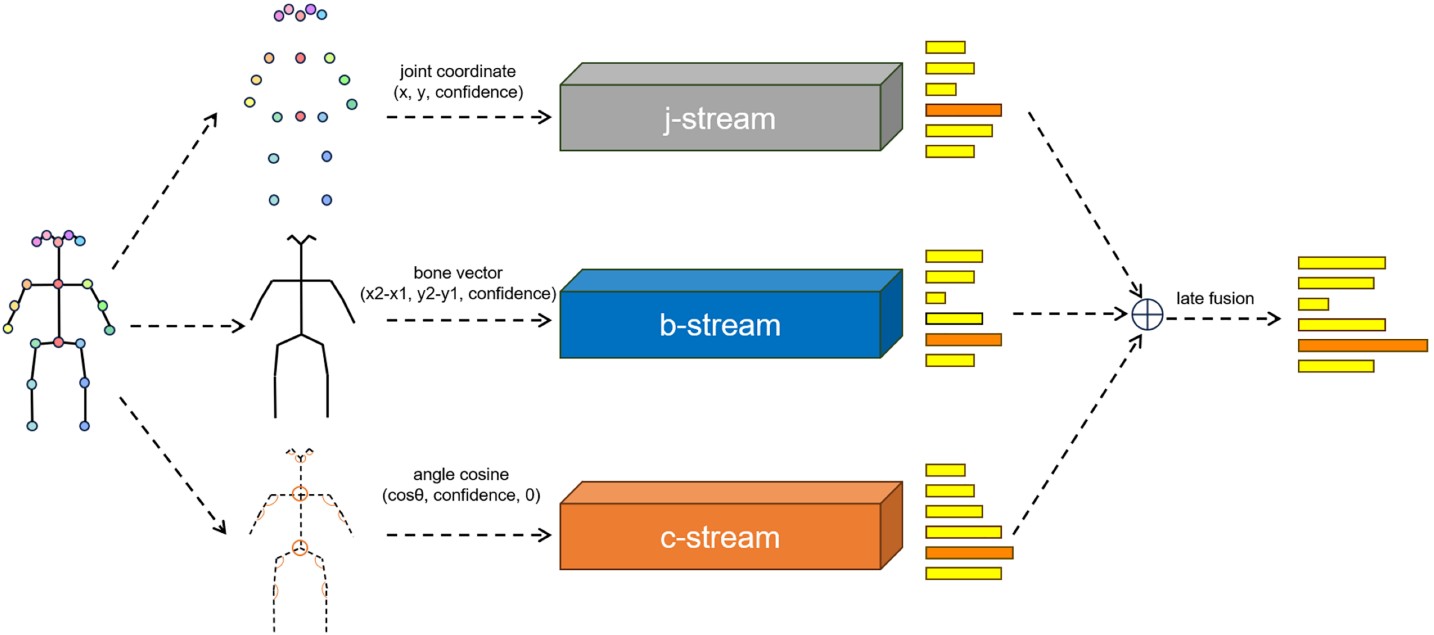

**Figure 2  Illustration of the overall archtecture of the network.** The softmax scores of the three streams are fused using weighted summation to obtain the final prediction. j denoted the joint information. b denotes the bone information. c denotes the angle information.

(1) Sort video frames based on the keyframe selection indicator. Choose frames with the smallest indicator to create a new subsequence of M frames, appending it to the N-frame subsequence from Uniform Sampling.

(2) Divide the sequence into M non-overlapping substrings. Select the frame with the smallest indicator from each substring to form a new M-frame subsequence. Connect it to the N-frame subsequence from Uniform Sampling.

(3) Building upon (1), rearrange the generated N+M frames chronologically to create a new downsampling sequence.

(4) Building upon (2), rearrange the generated N+M frames chronologically to create a new downsampling sequence.

As illustrated in the Fig. 3, the blue section represents the uniform sampling process, which evenly divides the total frames into M non-overlapping segments, randomly samples frames within each segment, and combines the M selected frames into a subsequence. The orange section represents the process of downsampling keyframes, where the depth of the orange color indicates the intensity of the indicator: the darker the color, the smaller the indicator, signifying a greater degree of joint opening. The difference between Strategy (1) and Strategy (2) lies in the keyframe selection method. Strategy (1) directly selects N frames to form a subsequence based on the intensity of the indicator, while Strategy (2) evenly divides the total frames into N non-overlapping segments in the time dimension, selects the frame with the highest indicator intensity within each segment,

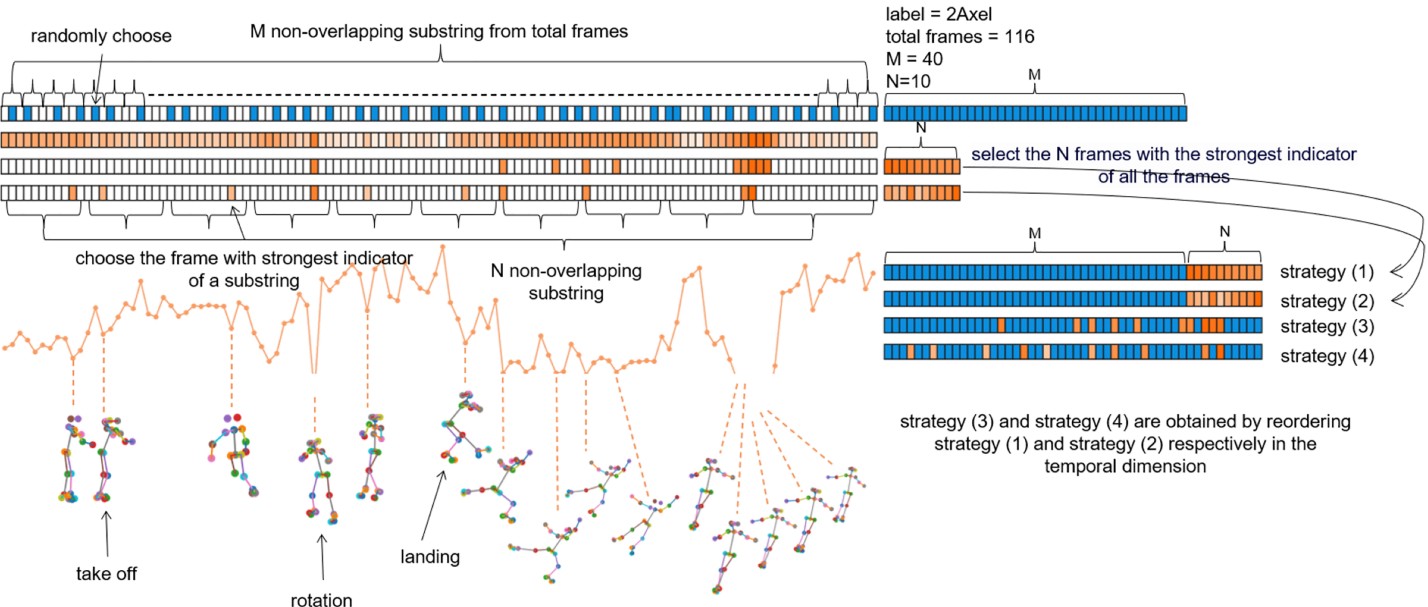

**Figure 3** **Illustration of Uniform Sampling and Keyframe Sampling.** The blue grids represents uniform sampling, while the orange grids depicts keyframe sampling. In this sample, the total number of frames is 116, with an assumed M of 40 and N of 10.

and forms these N frames into a subsequence. The main distinction between Strategies (1) and (2) *vs.* Strategies (3) and (4) is in how the M-frame and N-frame subsequences are combined. Strategies (1) and (2) simply append the N frames after the M frames, while Strategies (3) and (4) add a reordering step, aiming to maintain the temporal continuity of the samples. We will further analyze these different strategies in the ablation study section.

# EXPERIMENTS

## Datasets

**FSD-10.** The Figure Skating Dataset (FSD-10) (*Liu et al., 2020a*) is a challenging dataset in competitive sports, featuring 1,484 figure skating videos labeled with 10 actions. It includes 989 training and 495 testing videos, segmented from around 80 h of global figure skating championships (2017–2018).

**FineGYM.** FineGYM (*Shao et al., 2020*) is a high-quality action recognition dataset with 29 k videos and 99 fine-grained gymnastic action classes. Human poses are extracted using GT bounding boxes (provided by *Duan et al. (2022c)*).

**NTU RGB+D.** The NTU RGB+D dataset (*Shahroudy et al., 2016*) is a human action recognition dataset with 56,880 skeleton sequences from 40 volunteers, categorized into 60 classes. It suggests two evaluation setups: (1) Cross-subject (X-sub), with training from 20 subjects and testing from the remaining 20; (2) Cross-view (X-view), training from views 2 and 3, and testing exclusively from view 1. In our experiments, 2D human poses are estimated using HRNet (*Sun et al., 2019*) (provided by *Duan et al. (2022b)*).

## Implementation details

All experiments are conducted on one RTX 3090 GPU with PYSKL (*Duan et al., 2022b*) and MMaction2 (*Contributors, 2020*). Additionally, keyframe sampling was used to sample 25 frames in both datasets. By default, all models are trained with SGD with momentum 0.9, weight decay $5e^{-4}$. We set the batch size to 16, set the initial learning rate to 0.1. For the FSD-10 dataset, we repeated it 50 times, train the models for 64 epochs with CosineAnnealing learning rate scheduler. We use uniform sampling to sample 100 frames and keyframe sampling to sample 25 frames from skeleton data to form training samples. Previously, bone vectors were calculated by subtracting key point coordinates in the pipeline. Our proposed method uses cosine similarity based on these bone vectors. To improve efficiency, we suggest precomputing and storing both the bone vectors and cosine similarities.

## Experiment result

We conducted experiments on the FSD-10 dataset using CTRGCN as the baseline, and reported in Table 1 the improvements in various categories and mean accuracy after adding cosine stream and Keyframe Sampling.

FSD-10 is a fine-grained figure skating dataset where movements are categorized into spins, sequences, and jumps. Spins are further classified based on rotation posture, such as FlyCamelSpin4 and ChComboSpin4, which are both level four spins (though levels are not distinguished in this dataset). Sequences include two types of footwork: StepSequence, which combines several defined movements and can be divided into four levels, and ChoreSequence, which is more flexible and not graded. For jumps, there are six basic types —Axel, Toeloop, Flip, Lutz, Salchow, and Loop—categorized by the entry action. Jumps are further classified by the number of rotations, and combinations to form combinations jumps or sequence jump. In FSD-10, examples include 2Axel, 3Loop, 3Flip, 3Axel, 3Lutz, and 3Lutz_3Toeloop, with the latter being a combinations jump.

As shown in Table 1, the main categories with improved accuracy after introducing the cosine stream and Keyframe Sampling are ChoreSequence1, StepSequence3, 3Flip, and 3Lutz. The improvement in sequence accuracy aligns with our expectations, as sequences are composed of a series of sub-actions. Compared to other types of actions, sequences often have longer durations and noticeable displacement effects. Introducing angle features helps mitigate the impact of displacement. The improvements in Flip and Lutz accuracy are also expected. As shown in Fig. 1 right side, these two actions have very high similarity and are often the most controversial in referee decisions. The introduction of angle features effectively helps the model distinguish the subtle differences between these fine-grained actions. The improvement in 3Loop accuracy may be due to training errors, and returned to normal values when Keyframe Sampling was introduced.

## Ablation study

In this section, we analyze the proposed cosine stream and Keyframe Sampling algorithm on the FSD-10 dataset. For the cosine stream, we selected three latest and widely recognized skeleton-based action recognition models—AGCN (*Shi et al., 2019*), MSG3D

**Table 1 Comparison of accuracy of various categories using cosine stream and Keyframe Sampling in CTRGCN in FSD-10 dataset.** jb denotes joint and bone two streams, jbc denotes adding cosine stream. uni denotes original uniform sampling, key denotes ours keyframe sampling.

| | ChComboSpin4 | 2Axel | ChoreoSequence1 | 3Loop | StepSequence3 |
|---|---|---|---|---|---|
| CTRGCN_jb_uni | 100.0 | 93.6 | 75.8 | 95.4 | 88.9 |
| CTRGCN_jbc_uni | 100.0 | 93.6 | **80.7**↑ | **97.7**↑ | 88.9 |
| CTRGCN_jbc_key | 100.0 | **95.7**↑ | **83.9**↑ | 95.4 | **91.7**↑ |

| | 3Flip | FlyCamelSpin4 | 3Axel | 3Lutz | 3Lutz_3Toeloop | Mean accuracy |
|---|---|---|---|---|---|---|
| CTRGCN_jb_uni | 90.9 | 97.5 | 90.9 | 94.1 | 96.4 | 92.5 |
| CTRGCN_jbc_uni | **95.2**↑ | 97.5 | 90.9 | 94.1 | 96.4 | **93.5**↑ |
| CTRGCN_jbc_key | **97.6**↑ | 97.5 | 90.9 | **97.1**↑ | 96.4 | **94.6**↑ |

(*Liu et al., 2020b*), and CTRGCN (*Chen et al., 2021*)—as baselines. No modifications are required to the network structure for the cosine stream. For the Keyframe Sampling algorithm, we chose the current state-of-the-art model CTRGCN (*Chen et al., 2021*) as the baseline and demonstrated its effectiveness on the joint, bone, and cosine streams. As publicly available results for these methods on the FSD-10 dataset were not found, we conducted our experiments on this dataset using networks successfully reproduced from the PYSKL (*Duan et al., 2022b*) toolbox. The experiments were conducted with the same hyper-parameter settings to ensure fairness and consistency in the evaluation.

**Effectiveness of cosine stream.** We also evaluated the cosine stream on three widely used skeleton-based methods (Table 2). Despite slightly lower performance compared to joint and bone streams, the cosine stream employs 1D cosine similarity data, while the others use 2D coordinate data. However, the aggregated three-stream model consistently outperforms two-stream methods in Mean Class Accuracy and Top1 Accuracy. The addition of cosine stream improves Mean Class Accuracy by 0.9%, and Top1 Accuracy by 0.7% in AGCN, by 0.8% and 0.2% in MSG3D, and by 1.0% and 1.2% in CTRGCN, respectively. This proves the effectiveness of cosine stream.

In addition, we conducted supplementary experiments on the latest InfoGCN and HD-GCN, with results presented in Table 3. Since both models are enhanced versions of AGCN and CTRGCN, they include numerous attention modules, leading to increased model complexity. Given the limited amount of FSD-10 data, this complexity resulted in overfitting, even though we employed more data augmentation techniques than in the original work. Nevertheless, the introduction of cosine flow still yielded a 1.2% to 1.4% accuracy improvement, demonstrating its effectiveness.

Due to the prevalent use of motion features in current state-of-the-art methods, we incorporated experiments involving joint motion, bone motion, and our proposed cosine stream extended to include cosine motion in the temporal dimension in our experiments with CTRGCN. As shown in Table 4, we observed that both joint motion and bone motion, as well as cosine motion, performed worse than the spatial dimension feature streams. This indicates that the motion information performs poorly when faced with situations where the correlation between speed and action itself is not significant, which is

**Table 2 Improvement of the Cosine Stream Across Different Models on FSD-10 Dataset.** Here, j and b represent the joint stream and bone stream, respectively, while c signifies our proposed cosine stream.

| Acc (%) | AGCN | | | | |
| --- | --- | --- | --- | --- | --- |
| | j | b | c | j&b | j&b&c |
| Mean class | 90.2% | 91.5% | 87.7% | 91.7% | **92.6%↑** |
| Top1 | 88.9% | 90.4% | 87.1% | 90.8% | **91.5%↑** |
| Acc (%) | MSG3D | | | | |
| | j | b | c | j&b | j&b&c |
| Mean class | 90.1% | 90.5% | 89.4% | 90.9% | **91.7%↑** |
| Top1 | 90.1% | 89.2% | 88.5% | 90.6% | **90.8%↑** |
| Acc (%) | CTRGCN | | | | |
| | j | b | c | j&b | j&b&c |
| Mean class | 90.9% | 91.5% | 90.1% | 92.5% | **93.5%↑** |
| Top1 | 90.1% | 90.8% | 90.4% | 92.0% | **93.2%↑** |

**Table 3 Improvement of the Cosine Stream Across InfoGCN and HD-GCN on FSD-10 dataset.**

| Model | InfoGCN | | | | |
| --- | --- | --- | --- | --- | --- |
| Stream | j | b | c | j&b | j&b&c |
| Top1 acc (%) | 82.4 | 82.8 | 81.8 | 85.4 | **86.6↑** |

| Model | HD-GCN | | | | | | | |
| --- | --- | --- | --- | --- | --- | --- | --- | --- |
| Stream | j_com_1 | b_com_1 | c_com_1 | j_com_8 | b_com_8 | c_com_8 | j&b | j&b&c |
| Top1 acc (%) | 83.3 | 82.1 | 80.8 | 82.4 | 83.1 | 81.1 | 88.5 | **89.9↑** |

**Table 4 Improvement of the Cosine Stream (c) and Cosine Motion Stream (cm) with CTRGCN on the FSD-10 Dataset, where j, b, jm, and bm denote the joint stream, bone stream, joint motion stream, and bone motion stream, respectively.**

| Acc (%) | CTRGCN | | | | | |
| --- | --- | --- | --- | --- | --- | --- |
| | j | jm | b | bm | c | cm |
| Mean class | 90.9 | 89.5 | 91.5 | 89.6 | 90.1 | 88.6 |
| Top1 | 90.1 | 88.2 | 90.8 | 88.7 | 90.4 | 87.3 |

| Acc (%) | CTRGCN | | | |
| --- | --- | --- | --- | --- |
| | j&b | j&b&jm&bm | j&b&c | j&b&jm&bm&c&cm |
| Mean class | 92.5 | 93.2 | **93.5↑** | **93.6↑** |
| Top1 | 92.0 | 92.5 | **93.2↑** | 93.2 |

consistent with our expectations. Furthermore, the fusion of joint motion and bone motion with the original joint-bone dual-stream model (j&b) only resulted in a modest increase of 0.7% in Mean Class accuracy and 0.5% in Top1 accuracy, while the inclusion of cosine

**Table 5 Results of CTRGCN on the FSD-10 dataset, showcasing various Keyframe Sampling strategies for each stream.** Origin represents no keyframe sampling is applied, while uniform sampling selects frames at a fixed interval N. Numbers 0-4 correspond to the specific keyframe sampling strategies described in the subsection on Keyframe Sampling algorithm.

| Acc (%) | Strategies for joint stream | | | | | |
|---|---|---|---|---|---|---|
| | j_origin | j0 | j1 | j2 | j3 | j4 |
| Mean class | 90.9 | 91.4 | 89.4 | 90.7 | **91.6**↑ | **92.4**↑ |
| Top1 | 90.1 | 90.1 | 88.2 | **90.6**↑ | **90.4**↑ | **91.5**↑ |
| Acc (%) | Strategies for bone stream | | | | | |
| | b_origin | b0 | b1 | b2 | b3 | b4 |
| Mean class | 91.5 | 92.1 | 89.8 | 91.7 | **92.9**↑ | **94.3**↑ |
| Top1 | 90.8 | 91.3 | 89.9 | 91.1 | **92.7**↑ | **93.2**↑ |
| Acc (%) | Strategies for cosine stream | | | | | |
| | c_origin | c0 | c1 | c2 | c3 | c4 |
| Mean class | 90.1 | 89.4 | 89.5 | **90.9**↑ | 90.1 | **90.6**↑ |
| Top1 | 90.4 | 88.7 | 88.7 | 90.4 | 88.7 | 89.4 |
| Acc (%) | multi-stream | | | | | |
| | j0&b0 | j4&b4 | | j4&b4&c4 | | j4&b4&c_origin |
| Mean class | 92.5 | **94.3**↑ | | **94.3**↑ | | **94.6**↑ |
| Top1 | 91.8 | **93.2**↑ | | **93.2**↑ | | **94.4**↑ |

motion brought about negligible improvement. We hypothesize that this may be due to the fact that the actions in the dataset generally involve certain speeds during execution, which are not significantly correlated with the actions themselves, thereby resulting in unsatisfactory performance of the motion features. Therefore, we did not utilize motion features in subsequent experiments.

**Effectiveness of keyframe sampling.** We explored various keyframe sampling and uniform sampling strategies 3, with results shown in Table 5. Strategy analysis reveals that merely increasing downsampled frames (Origin and Strategy (0) columns in the prior three sub tables) may lead to a slight improvement in model performance. Strategies (1) through (4) share the same frame rate as Strategy (0), with the exception that the M frames are obtained through keyframe sampling. As depicted in the chart, the performance differences stem from how these frames are processed.

- Strategy (1): Simply concatenates the strongest M frames based on an indicator after the N-frame subsequences. However, this disrupts the temporal continuity of the actions since the strongest frames may not be sequential, resulting in a slight decrease in accuracy.
- Strategy (3): After selecting the M keyframe frames, it immediately downsamples and reorders both the keyframes and N-frame subsequences in the temporal dimension to ensure continuity. This approach significantly improves accuracy.

- Strategy (2): Demonstrates another method of keyframe selection by dividing the video into M non-overlapping temporal segments and selecting the frame with the strongest indicator from each segment to form a new subsequence. This method naturally maintains temporal continuity and has a more even distribution, leading to better performance compared to Strategy (1).
- Strategy (4): Goes a step further by rearranging the entire M+N frame samples. This finer adjustment enhances the assistance provided to the model, resulting in further improvements.

In summary, the key lies in effectively selecting and utilizing keyframes while maintaining temporal continuity and even distribution. Strategies (3) and (4) achieve this through more sophisticated processing, thereby boosting the model's accuracy. Compared with origin strategy, Strategy (4) notably improves mean class accuracy by 1.5% and Top1 Accuracy by 1.4% in the joint stream, and by 2.8% and 2.4% in the bone stream, respectively. If compared with Strategy (1), the improvements in the joint stream are 1.0% and 1.4%, and the improvements in the bone stream are 2.2% and 1.9%. This proves the effectiveness of Keyframe Sampling and can help improve the performance of the model under appropriate combination strategies.

Analyzing the third sub-table reveals keyframe sampling does not improve the cosine stream. This is attributed to the indicator using raw data from the cosine stream, causing redundancy and a slight performance decrease. This confirms that improvements in the first two sub-tables are due to keyframes rather than increased downsampled frames.

Fusing the streams with keyframe sampling (fourth sub-table) involves Strategy (4) for joint and bone streams. Compared to the original joint-bone two-stream model, there is a 1.8% improvement in mean class accuracy and 1.2% in Top1 Accuracy. However, keyframe sampling does not enhance the performance of the cosine stream. On the other hand, when fused with the cosine stream using the original strategy, it results in a 2.1% increase in mean class accuracy and 2.4% in Top1 Accuracy. This proves that keyframe sampling does not conflict with the introduction of cosine stream in improving model performance.

## Cross-dataset validations

To validate the applicability of our method, we conducted cross-dataset validation on the FineGYM and NTU RGB+D datasets, in addition to the experiments conducted on FSD-10 as mentioned above.

Table 6 presents the results of introducing the cosine stream on the FineGYM dataset, showcasing a 0.5% improvement in mean class accuracy and a 0.2% improvement in Top-1 Accuracy for the original joint-bone two-stream model. When employing keyframe sampling strategy four for both the joint stream and bone stream, the three-stream model exhibited a 1.0% increase in mean class accuracy and a 0.6% increase in Top-1 Accuracy compared to the original joint-bone two-stream model. The observed smaller enhancement is attributed to dataset differences, where the categorization of action classes in the FineGYM dataset may have less correlation with the angular relationships of joints within the body.

**Table 6 Enhanced CTRGCN model performance on FineGYM dataset with cosine stream and Keyframe Sampling.**

| Acc (%) | CTRGCN | | | | |
|---|---|---|---|---|---|
| | j | b | c | j&b | j&b&c |
| Mean class | 88.7 | 91.4 | 85.0 | 92.0 | 92.5↑ |
| Top1 | 91.9 | 93.7 | 89.0 | 94.5 | 94.7↑ |
| Acc (%) | CTRGCN & Keyframe Sampling | | | | |
| | j4 | b4 | c4 | j4&b4 | j4&b4&c |
| Mean class | 89.5↑ | 91.5↑ | 84.1 | 92.6↑ | 93.0↑ |
| Top1 | 92.6↑ | 93.9↑ | 88.4 | 94.8↑ | 95.1↑ |

**Table 7 CTRGCN model performance on NTU RGB+D dataset with cosine stream and Keyframe Sampling.**

| Acc (%) | CTRGCN on X-sub | | | | |
|---|---|---|---|---|---|
| | j | b | c | j&b | j&b&c |
| Top1 | 89.3 | 91.6 | 87.8 | 92.3 | 92.9↑ |
| Acc (%) | CTRGCN & Keyframe Sampling on X-sub | | | | |
| | j | b | c | j&b | j&b&c |
| Top1 | 89.9 | 91.4 | 87.3 | 92.5 | 92.8 |
| Acc (%) | CTRGCN on X-view | | | | |
| | j | b | c | j&b | j&b&c |
| Top1 | 96.2 | 96.1 | 87.0 | 97.3 | 97.4↑ |
| Acc (%) | CTRGCN & Keyframe Sampling on X-view | | | | |
| | j | b | c | j&b | j&b&c |
| Top1 | 95.7 | 95.7 | 87.5 | 97.2 | 97.2 |

In Table 7, the results indicate that the enhancements of the model with the cosine stream and Keyframe Sampling on the NTU RGB+D dataset are not as promising. This holds true for both our experimental results and the data results cited in other works. We attribute this observation to two factors: (1) In terms of the inherent categorization of the dataset, NTU RGB+D places less emphasis on human joint angles. (2) The dataset is collected in a controlled lab environment where subjects do not exhibit significant displacement relative to the camera, leading to a loss of the advantageous shift, scale, and rotation invariance of angle features.

## LIMITATIONS
Our method excels at handling the 'near large, far small' phenomenon that occurs during 3D-to-2D projection, as well as actions involving significant character displacement within an image. However, it has a limitation: the cosine features we use are limited to two

**Table 8 CTRGCN model performance on NTU RGB+D_3D dataset with cosine stream and Keyframe Sampling.** KS denotes using keyframe sampling.

| Acc (%) | CTRGCN on X-sub-3d | | | | |
|---|---|---|---|---|---|
| | j | b | c | j&b | j&b&c |
| Top1 | 89.6 | 90.0 | 80.9 | 91.5 | 91.5 |
| KS Top1 | 89.9 | 90.0 | 80.5 | 91.8 | 91.9 |
| Acc (%) | CTRGCN on X-view-3d | | | | |
| | j | b | c | j&b | j&b&c |
| Top1 | 95.6 | 95.4 | 84.4 | 96.6 | 96.6 |
| KS Top1 | 95.5 | 95.5 | 84.6 | 96.7 | 96.7 |

dimensions—cosine similarity and confidence—which may not be as expressive as the three-dimensional keypoint coordinates and skeleton vectors in a single stream.

Although we achieved good results on the FSD-10 dataset, this success stems from a careful balancing of our method's strengths and weaknesses. In scenarios where actions involve minimal displacement or where the raw data is inherently 3D, our method may be less effective. For instance, in the X-view of the NTU RGB+D dataset (Table 7), the cross-view enhancements of the original data or the data's inherent 3D nature (Table 8) mean that depth information is not lost during 3D-to-2D projection. Consequently, the advantages of the cosine stream are not fully realized, while its limitations persist, resulting in significantly lower performance compared to the other two streams.

Addressing the challenge of expanding cosine representation to mitigate its two-dimensional limitation is an important direction for future research.

## CONCLUSION

In this work, we propose a cosine stream that leverages joint angles and the cosine similarity of bone vectors for accurate action prediction. Additionally, we introduce a keyframe sampling method based on joint cosine values to quantify body extension. Our approach excels in handling 2D data by mitigating the loss of depth information during 3D-to-2D projection, particularly for movements involving significant displacement.

The main contribution of this research lies in providing an effective method for action recognition, especially in preserving spatial information during the projection from 3D to 2D. Our method demonstrates strong robustness in handling complex actions with large displacements. This achievement not only extends the current body of knowledge in skeleton-based action recognition but also lays the foundation for future studies.

However, the work also has limitations. The cosine features are confined to two dimensions—cosine similarity and confidence—making them less expressive than three-dimensional keypoint coordinates and skeleton vectors. Future research could explore ways to expand the representation capabilities of the cosine stream to overcome its

two-dimensional limitation. Additionally, improving the keyframe sampling method to enhance adaptability and generalization is another important direction for further investigation.

Overall, this work provides a novel perspective and technical approach to action recognition, with practical implications particularly in the context of 3D-to-2D projection. Future work can focus on extending cosine representation, refining the sampling method, and exploring broader applications across other datasets, thereby advancing the field further.

### Funding
This work was supported by the National Key R&D program of China (2020YFB1712401), the Nature Science Foundation of China (62006210, 62206252), the Key science and technology project of Henan province (221100211200, 221100210100), the technological research projects in Henan province (232102210090). The funders had no role in study design, data collection and analysis, decision to publish, or preparation of the manuscript.

### Grant Disclosures
The following grant information was disclosed by the authors:
National Key R&D Program of China: 2020YFB1712401.
Nature Science Foundation of China: 62006210, 62206252.
Key Science and Technology Project of Henan Province: 221100211200, 221100210100.
Technological Research Projects in Henan Province: 232102210090.

### Competing Interests
Yidan Chen is employed by Shangu Cyber Security Technology Company Limited.

### Author Contributions
- Chengming Liu conceived and designed the experiments, analyzed the data, performed the computation work, authored or reviewed drafts of the article, and approved the final draft.
- Jiahao Guan conceived and designed the experiments, performed the experiments, analyzed the data, performed the computation work, prepared figures and/or tables, authored or reviewed drafts of the article, and approved the final draft.
- Haibo Pang analyzed the data, authored or reviewed drafts of the article, and approved the final draft.
- Lei Shi analyzed the data, authored or reviewed drafts of the article, and approved the final draft.
- Yidan Chen analyzed the data, authored or reviewed drafts of the article, and approved the final draft.

## Data Availability

The code and data are available at GitHub and Zenodo:

- https://github.com/Jiahao-Guan/pyskl_cosine

- Jiahao Guan. (2024). Jiahao-Guan/pyskl_cosine: v0.1 (first). Zenodo. https://doi.org/10.5281/zenodo.13679682.

The FSD-10 dataset is available at GitHub: https://shenglanliu.github.io/fsd10.

The FineGym and NTU RGBD 60 datasets are available at Zenodo: Guan. (2024). gym_nturgbd60 [Data set]. Zenodo. https://doi.org/10.5281/zenodo.14049728.

## Supplemental Information

Supplemental information for this article can be found online at http://dx.doi.org/10.7717/peerj-cs.2523#supplemental-information.

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
