# Peer review of "Angle information assisting skeleton-based actions recognition"

_PeerJ Computer Science, doi:10.7717/peerj-cs.2523_

## Round 0.1 · original submission · Major Revisions

Dear authors,

Thank you for submitting your article. Based on reviews' comments, your article has not yet been recommended for publication in its current form. However, we encourage you to address the concerns and criticisms of the reviewer and to resubmit your article once you have updated it accordingly. Before submitting the paper following should also be addressed:

1. In general, the literature review is not sufficient. More recent literature should be explored in depth. It is more of the type “researcher X did Y” rather than an authoritative synthesis assessing the current state-of-the-art. Advantages and disadvantages of the related works should be evaluated.
2. pros and cons of the methods should be clarified. What are the limitation(s) methodology(ies) adopted in this work? Please indicate practical advantages, and discuss research limitations.
3. Some paragraphs are too long to read. Long paragraphs should be divided into two or more for readability and comprehensibility.
4. The space character should be used correctly in its place.
5. The conclusion section is indicative, but it might be strengthened to highlight the importance and applicability of the work done with some more in-depth considerations, to summarize the findings, and to give readers a point of reference. Additional comments about the reached results should be included.

Best wishes,

Reviewer 1 ·

Basic reporting

(1) Overall Writing and Presentation:
- The manuscript is generally well-written, but it contains minor typographical errors (e.g., citation markers in Lines 38 and 91, single quotes in captions for Tables 1, 2, and 3).
- There are too many and overly large figures related to motivation (Figures 1 and 2). Conversely, there are insufficient figures to help understand the key proposed methodologies, such as the Cosine stream and keyframe downsampling algorithm. Relocating Figure 3 to the "Effectiveness of Keyframe Sampling" section within the Experiments would be appropriate.
- The paper lacks a detailed explanation of the proposed keyframe-based downsampling, making the methodology difficult to comprehend (Lines 114-124).
- Consistent terminology usage is necessary (e.g., Line 99: Keyframe downsampling, Line 113: keyframe-based downsampling, Line 176: Keyframesampling).

Experimental design

(1) Scope of Experiments:
- The manuscript handles only pre-2021 GCN-based methods. Additional experiments involving more recent GCN-based methods are required. Suggested papers include: "InfoGCN: Representation Learning for Human Skeleton-based Action Recognition (CVPR 2022)", "Hierarchically Decomposed Graph Convolutional Networks for Skeleton-Based Action Recognition (ICCV 2023)", and "BlockGCN: Redefine Topology Awareness for Skeleton-Based Action Recognition (CVPR 2024)".

(2) Class-wise Performance Analysis:
- A detailed analysis of class-wise performance improvements due to the addition of the Cosine stream is necessary.

(3) Sampling Visualization:
- Additional visualizations of frames sampled in various scenarios are needed in Figure 3.

(4) Accuracy Table Issues:
- Table 5 contains redundant entries for Top1 Acc(%). Furthermore, the performance of the c stream in CTRGCN on X-view is significantly lower compared to the j and b streams, which seems unusual and requires clarification.

Validity of the findings

(1) Novelty:
- The novelty of the proposed approach is insufficient. Prior works like "Fusing Higher-Order Features in Graph Neural Networks for Skeleton-Based Action Recognition (TNNLS 2022)" and "Shifting Perspective to See Difference: A Novel Multi-View Method for Skeleton based Action Recognition (arXiv 2022)" also utilize angular features and angle representation. Emphasizing the differences from these methods is necessary.

(2) Cosine Stream Concerns:
- The proposed Cosine stream measures the angle between bones. However, the datasets used for experiments involve 2D skeleton data extracted by pose estimation networks, where 3D skeletons are projected onto the image plane. This raises concerns about the meaningfulness of angular features. Visualizing the cosine stream input for 2D skeleton data and additional experiments and analysis on 3D skeleton data are needed.

Additional comments

(1) Improvements and Additions:
- Correct typographical errors, conduct additional experiments with 3D skeleton data for the Cosine stream, provide detailed analysis and insightful visualizations of the methodology, and emphasize the novelty of the proposed approach.

Cite this review as

·

Basic reporting

No comments

Experimental design

No comments

Validity of the findings

No comments

Additional comments

- The abstract is quite detailed and informative, but it could benefit from being more concise. Focus on the main contributions and results, and try to reduce redundancy. For example, you mention the datasets and the improvement in accuracy twice.
- The introduction covers a wide range of topics, from the general context of action recognition to specific challenges in figure skating. It would be beneficial to organize this section more logically, starting with the broader context, then narrowing down to the specific challenges, and finally introducing your proposed solution.
- While the related work section covers several key papers, it could be expanded to include more recent advancements in the field. Additionally, providing a comparative analysis of these works in relation to your method would help highlight the novelty and significance of your contributions.
- The explanation of the cosine stream and keyframe downsampling algorithm is clear, but some parts could benefit from additional detail. For example, the mathematical formulation of the cosine similarity could be expanded with more context on why this metric was chosen and how it compares to other potential metrics.
- The figures provided are helpful, but their captions could be more descriptive. Ensure that each figure is fully explained in the text and that the captions provide enough context for readers to understand the figure without having to refer back to the text.
- While you provide some details on the experimental setup, including the hardware and software used, more information on the training parameters, such as learning rates, batch sizes, and the number of epochs, would enhance reproducibility. Additionally, a discussion on the computational complexity of your method compared to others would be valuable.
- The results section provides quantitative improvements, but it would be beneficial to include a more in-depth analysis of why your method outperforms others. Discuss the specific scenarios or types of actions where your method excels and any limitations or failure cases observed.
- While you conducted ablation studies, the details provided are minimal. Expand on these studies by including more variations of your method and discussing the impact of each component in greater detail.
- The conclusion should summarize the key findings more succinctly and provide a clear outline of potential future work. Mention specific directions for further research and how your method could be extended or applied to other domains.

Cite this review as

---

## Round 0.2 · Minor Revisions

Dear authors,

Thank you for submitting your paper. Although you appear to have attempted to address the reviewers' comments, you have not addressed the editor's concerns and criticisms. We encourage you to make these necessary additions and changes and resubmit your article once you have updated it accordingly. The following concerns and criticisms are resent to you for your consideration:

1. In general, the literature review is not sufficient. More recent literature should be explored in depth. It is more of the type “researcher X did Y” rather than an authoritative synthesis assessing the current state-of-the-art. Advantages and disadvantages of the related works should be evaluated.
2. pros and cons of the methods should be clarified. What are the limitation(s) methodology(ies) adopted in this work? Please indicate practical advantages, and discuss research limitations.
3. Some paragraphs are too long to read. Long paragraphs should be divided into two or more for readability and comprehensibility.
4. The space character should be used correctly in its place.
5. The conclusion section is indicative, but it might be strengthened to highlight the importance and applicability of the work done with some more in-depth considerations, to summarize the findings, and to give readers a point of reference. Additional comments about the reached results should be included.

Best wishes,

Reviewer 1 ·

Basic reporting

The rebuttal addresses most of the major concerns from the reviewers and highlights significant improvements made to the manuscript. The responses show that typographical issues were corrected, figures were reorganized and expanded for clarity, terminology was standardized, and additional experiments were conducted. They also added visualizations, clarified the novelty of their method, and improved the manuscript's overall organization. Given the comprehensive revisions, the paper seems well-positioned for final acceptance.

Experimental design

no comment

Validity of the findings

no comment

Cite this review as

---

## Round 0.3 · Minor Revisions

Dear author,

Thank you for the revision. Your paper needs a minor revision. The recommendations on Conclusion section seem to be not properly addressed. The conclusion section is still weak. The academic implications, main findings, shortcomings and directions for future research should be provided in the Conclusion section. What will be happen next? What should we expect from future papers? So rewrite it and consider the following comments:

- Highlight your analysis and reflect only the important points for the whole paper.
- Mention the benefits
- Mention the implication in the last of this section.

Best wishes,

---

## Round 0.4 · accepted · Accept

Dear Authors,

I am grateful to you for responding in such a forthright manner to the concerns and criticism raised by the reviewers and editor. I am satisfied that the paper has been sufficiently improved and is now acceptable for publication.

Best wishes,

·

Basic reporting

No further comments

Experimental design

No further comments

Validity of the findings

No further comments

Additional comments

No further comments

Cite this review as